# Galectin-3, N-terminal Propeptides of Type I and III Procollagen in Patients with Atrial Fibrillation and Metabolic Syndrome

**DOI:** 10.3390/ijms21165689

**Published:** 2020-08-08

**Authors:** Valery A. Ionin, Elena I. Baranova, Ekaterina L. Zaslavskaya, Elena Yu. Petrishcheva, Aleksandr N. Morozov, Evgeny V. Shlyakhto

**Affiliations:** 1Almazov National Medical Research Centre, Institute of Endocrinology, Metabolic Syndrome Laboratory, 197341 St. Petersburg, Russia; baranova.grant2015@yandex.ru (E.I.B.); elenasopilova@mail.ru (E.Y.P.); e.schlyakhto@almazovcentre.ru (E.V.S.); 2Department of Internal Diseases №1, Department of Surgery №2, Pavlov University, 197022 St. Petersburg, Russia; Memlikster@gmail.com (E.L.Z.); sanmor@mail.ru (A.N.M.)

**Keywords:** galectin-3, PINP, PIIINP, atrial fibrillation, metabolic syndrome, fibrosis

## Abstract

The aim of this study was to determine the concentration of galectin-3, PINP and PIIINP in patients with metabolic syndrome (MS) and atrial fibrillation (AF) with an assessment of the relationship with severity of left atrium fibrosis. A total of 480 subjects were included in the case-control study: MS patients (*n* = 337), 176 of whom had AF, 72 patients with AF without MS and 71 healthy subjects. Galectin-3, PINP and PIIINP blood concentrations and metabolic parameters were compared with the severity of left atrium fibrosis, measured by CARTO3. Galectin-3 in AF and MS patients is higher than in MS without AF and in healthy subjects (10.3 (4.8–15.4), 5.1 (4.3–8.8), 3.2 (2.4–4.2) ng/mL, *p* < 0.0001). Galectin-3 serum concentration in AF patients with MS is higher than in patients without MS: 10.3 (4.8–15.4), 6.8 (5.2–8.1) ng/mL, *p* = 0.0001. PINP and PIIINP concentration were higher in patients with AF and MS than in MS without AF: 3499.1 (2299.2–4567.3), 2130.9 (1425.3–2861.8) pg/mL, *p* < 0.0001, 94.9 (64.8–123.5), 57.6 (40.5–86.9) ng/mL, *p* < 0.0001. Galectin-3 correlates with PINP (r = 0.496, *p* < 0.001) and PIIINP concentration (*r* = 0.451, *p* < 0.0001). The correlation between galectin-3, PINP and the severity of left atrium fibrosis was found (r = 0.410, *p* < 0.001; *r* = 0.623, *p* < 0.001). Galectin-3 higher than 12.6 ng/mL increased the risk of AF more than five-fold. High galectin-3, PINP and PIIINP concentrations were associated with heart remodeling in MS patients and increased the risk of AF.

## 1. Introduction

Atrial fibrillation (AF), frequently occurring sustained supraventricular arrhythmia, is diagnosed in 1.5% of the adult population in developed countries of the world [1]. AF often leads to complications such as stroke, systemic thromboembolism and chronic heart failure [2]. In connection with the high prevalence and social importance of AF, continuing research into the causes, mechanisms of development and progression of this arrhythmia, the prevalence of AF, obesity and arterial hypertension have been steadily increasing in recent decades [3]. Epidemiological data largely explain the high prevalence of AF in young and middle-aged people without traditional causes of this arrhythmia (coronary artery disease, chronic heart failure, heart valve diseases) with abdominal obesity and hypertension, which are independent risk factors for AF [4]. It is known that the development of AF is based on various pathological conditions: hemodynamic disturbances, structural changes and atrial remodeling, electrical myocardial heterogeneity [5]. Cardiac remodeling including left ventricular hypertrophy, with the development of diastolic dysfunction and atrial dilatation is often detected in patients with abdominal obesity and hypertension. However, the regulatory mechanisms underlying myocardial changes are not fully understood at present. Currently, there is no doubt that myocardial fibrosis is the most important substrate for the development of AF [6]. It has been established that the severity of left atrial myocardial fibrosis negatively affects the progression of arrhythmia and the effectiveness of antiarrhythmic treatment [7]. Obesity and hypertension are the most common components of metabolic syndrome (MS), which can cause structural changes in the atria: an increased size of the left atrium and the development of interstitial atrial fibrosis [8]. There is a proposal that the severity of fibrosis indirectly can be evaluated by the level of galectin-3 in the blood, which is one of the markers of fibrosis. Galectin-3 is a protein from the family of β-galactoside binding proteins with a molecular weight of 30 kDa, which is an inducer of macrophage migration, as well as a powerful factor in the activation of fibroblasts and collagen synthesis, which is involved in the development of fibrosis. For the first time in 2004, Sharma et al., in an experimental study of hepatic fibrosis in mice, showed that galectin-3 expresses proliferating fibroblasts, and the administration of exogenous recombinant galectin-3 increases collagen production. These authors also demonstrated that galectin-3 in hypertrophied animal hearts was localized mainly around the nuclei of proliferating fibroblasts, while resting cells contained a minimal amount of galectin-3 in the cytoplasm [9]. Several studies have demonstrated that galectin-3 is capable of not only activating fibroblast proliferation, but also increasing the protein expression of α-smooth muscle actin (a marker of fibrosis intracellular) and extracellular proteins of collagen I and III types. In an animal model study, inhibition of galectin-3 was associated with inhibition of collagen production, collagen processing, degradation, cross-linking and deposition. Galectin-3 can also affect the destruction of the intercellular matrix through a system of tissue inhibitors of metalloproteinases and matrix metalloproteinases, thereby contributing to the molecular remodeling of the intercellular space [10]. The results of experimental work served as an opportunity to study the clinical significance of this biomarker of fibrosis in cardiovascular pathology. Several population-based studies have found that galectin-3 is associated with a risk of cardiovascular disease and it can be a predictor of heart failure and mortality [11]. In 2014, the first data was published demonstrating that patients with AF have a higher level of galectin-3 than in the general population. In particular, with a 10-year follow-up of 3306 participants in the Framingham study, it was found that 250 people (7.8%) had episodes of AF, while a higher level of circulating galectin-3 was associated with an increased risk of AF (OR = 1.19, 95% CI = 1.05–1.36, *p* = 0.03) [12]. Previously, Timonen demonstrated that in patients with dilated cardiomyopathy, the presence of AF was associated with a low N-terminal propeptide of procollagen I type (PINP) plasma level and an increased N-terminal propeptide of procollagen III type (PIIINP) in plasma. Therapy beta blockers and angiotensin-converting enzyme inhibitors (ACEi) reduced procollagen production, which seems promising in the treatment of patients with AF [13]. At the same time, there are practically no studies devoted to the investigation of the role of these circulating markers in the development of myocardial fibrosis in patients with MS and AF. Based on this, the purpose of this study was to establish the level of galectin-3 in serum and assess its relationship with procollagens and the severity of the myocardial fibrosis in patients with MS to determine the possible role of these factors in the occurrence of AF.

## 2. Results

The groups of patients were comparable in sex distribution and did not differ in age. The main clinical, laboratory and echocardiographic characteristics are presented in Table 1.

The concentration of galectin-3 in serum of patients with AF and MS is higher than in patients with MS without AF (10.3 (4.8–15.4) and 5.1 (4.3–8.8) ng/mL, *p* < 0.0001) and higher than in healthy subjects (10.3 (4.8–15.4) and 3.2 (2.4–4.2) ng/mL, *p* < 0.0001). In patients with MS without AF, galectin-3 concentration was also higher than in healthy subjects (5.1 (4.3–8.8) and 3.2 (2.4–4.2) ng/mL, *p* < 0.0001). When analyzing the data obtained in patients with AF, serum galectin-3 concentration results were obtained over a wide range, probably due to the influence of MS components. It was found that in patients with AF and the presence of MS, the concentration of galectin-3 in the serum is higher than in patients with AF without MS (10.3 (4.8–15.4) and 6.8 (5.2–8.1) ng/mL, *p* = 0.0001; Figure 1).

The concentration of PINP in plasma of patients with AF and MS is higher than in patients with MS without AF (3499.1 (2299.2–4567.3) and 2130.9 (1425.3–2861.8) pg/mL, *p* < 0.0001) and higher than in healthy subjects (3499.1 (2299.2–4567.3) and 1300.3 (800.1–2628.2) pg/mL, *p* < 0.0001). In patients with MS without AF the PINP concentration was also higher than in healthy subjects (2130.9 (1425.3–2861.8) and 1300.3 (800.1–2628.2) pg/mL, *p* = 0.008). It was found that in patients with AF and the presence of MS, the concentration of PINP in plasma is higher than in patients with AF without MS (3499.1 (2299.2–4567.3) and 2992.3 (2345.1–3663.1) ng/mL, *p* = 0.01; Figure 2).

The plasma concentration of PIIINP in patients with AF and MS is higher than in patients with MS without AF (94.9 (64.8–123.5) and 57.6 (40.5–86.9) ng/mL, *p* < 0.0001) and higher than in healthy subjects (94.9 (64.8–123.5) and 28.2 (20.9–36.5) ng/mL, *p* < 0.0001). In patients with MS without AF, the concentration of PIIINP was also higher than in healthy subjects (57.6 (40.5–86.9) and 28.2 (20.9–36.5) ng/mL, *p* = 0.001). It was established that in patients with AF and the presence of MS, the concentration of PIIINP in the blood serum is higher than in patients with AF without MS (94.9 (64.8–123.5) and 56.7 (46.9–64.5) ng/mL, *p* = 0.001; Figure 3).

According to the correlation analysis, galectin-3 relationship with anthropometric parameters characterizing obesity was identified in the general sample of the examined: body mass index (BMI; *r* = 0.352, *p* < 0.0001) and waist circumference (*r* = 0.362, *p* < 0.0001). Only PIIINP positively correlates with BMI (r = 0.305, *p* < 0.0001) and waist circumference (*r* = 0.315, *p* < 0.0001), but not the PINP. An analysis of the relationship with laboratory parameters found that galectin-3 correlates with the concentration of glucose (*r* = 0.315, *p* = 0.0001) and triglycerides in plasma (*r* = 0.264, *p* < 0.0001). A positive association of PIIINP with hypertriglyceridemia was detected (*r* = 0.343, *p* < 0.001). There were no significant correlations of galectin-3 and procollagens with total cholesterol and low-density lipoproteins (LDL) cholesterol levels in our study. According to the results of echocardiography, it was found that the concentration of galectin-3 in the blood positively correlates with the volume of the left atrium, which characterizes its dilatation (*r* = 0.426, *p* < 0.0001) and the volume of the right atrium (*r* = 0.341, *p* < 0.0001). Regression analysis also confirmed the association of galectin-3 with the volume of the left atrium (β = 0.177, *p* = 0.03) and the volume of the right atrium (β = 0.281, *p* = 0.001). Procollagen also positively correlated with the volume of the left atrium: PINP (*r* = 0.313, *p* < 0.0001) and the PIIINP (*r* = 0.315, *p* < 0.0001).

When evaluating the correlation of biomarkers circulating in the blood of patients with AF who underwent electroanatomical mapping before a radiofrequency pulmonary vein isolation (*n* = 68) there is a strong positive correlation of galectin-3 (*r* = 0,410, *p* < 0.001), PINP (*r* = 0.623, *p* < 0.001), with the severity of myocardial fibrosis of the left atrium and weak correlation with PIIINP (*r* = 0.256, *p* = 0.03), the concentration of galectin-3 in the blood positively correlated with the concentration of the PINP (*r* = 0.496, *p* < 0.0001) and the PIIINP (*r* = 0.451, *p* < 0.0001), as demonstrated in Figure 4.

We studied the risk of AF in the total cohort examined and in the cohort of patients with MS using receiver operating characteristic (ROC) analysis. We also determined the cut-off value concentrations of galectin-3, PINP and PIIINP, circulating in the blood, which can increase the risk of AF. In the total cohort of patients, the area under curve (AUC) of galectin-3 for prediction of AF (0.768 ± 0.03; 95% CI = 0.717–0.819) was not different from the AUC of PINP (0.761 ± 0.03, 95% CI = 0.708–0.813), but AUCs both of them were different significantly from AUC of PIIINP (0.707 ± 0.03, 95% CI = 0.651–0.763). In a cohort of MS patients, only the AUC of galectin-3 was higher and had a greater prognostic value in modeling the risk of AF. A cut-off value concentration of galectin-3 in patients with MS over 12.6 ng/mL increased the risk of AF by more than five-fold. In the cohort of patients with MS concentrations of PINP (≥3426.8 ng/mL) and PIIINP (≥56.7 pg/mL) increased the risk of AF in 8.9- and 4.4-fold, respectively. This data is presented in Table 2.

## 3. Discussion

The pathogenesis of AF is a multifactorial process, which is based on hemodynamic, structural, electrophysiological and molecular mechanisms. There are many clinical risk factors for AF, including old age, coronary artery disease, thyroid pathology, heart failure, alcohol abuse, chronic pulmonary pathology, etc. Among the patients examined in our study, these factors were excluded. In patients with MS, in contrast to the comparison group, three or more components of MS were diagnosed the most common among them were: hypertension, obesity, dyslipidemia. According to the Atherosclerosis Risk in Communities (ARIC) study, MS is a risk factor for AF [4]. In the basic mechanisms of structural myocardial remodeling, including atria, an important role belongs to aldosterone, as one of the components of the renin-angiotensin-aldosterone system that regulates blood pressure. The profibrogenic role of aldosterone has been known for a long time and is currently not in doubt [14]. Through the mechanism of development of fibrosis, as a possible substrate for nonvalvular AF, aldosterone, in turn, can act as a predictor of this rhythm disturbance. In experimental studies, it was found that aldosterone, through the activation of macrophages, promotes the production of active substances, including galectin-3, which increases the synthesis of type I and type III collagen by fibroblasts and leads to the development of myocardial fibrosis. In vitro and in vivo models, Lin et al. established PI3K/Akt, a signaling pathway for the induction of fibrosis by aldosterone via galectin-3 [15]. In our earlier study, we found that the concentration of aldosterone in patients with AF and MS is higher than in patients without arrhythmia, and the concentration of aldosterone positively correlates with galectin-3 [16].

The largest meta-analysis, which included 28 studies and 10813 patients, was published in 2020. This meta-analysis demonstrated that galectin-3 is higher in patients with AF than in patients with sinus rhythm, and an increased level of galectin-3 is associated with a higher risk of AF development (OR = 1.45, 95% CI = 1.15–1.83, *p* = 0.002) [17]. However, for many years the role of galectin-3 in the development of AF in patients with MS remained unknown. For the first time in 2016, in a pilot study, we published data that galectin-3 in patients with MS and AF is higher than in healthy control and it is associated with ineffective antiarrhythmic therapy [18]. At present, as a result of a larger study, the fact that galectin-3 in serum in patients with AF and MS is higher than in healthy subjects was confirmed, however, it was also found that this biomarker is higher in patients with AF and MS than in patients with MS without given arrhythmia. A feature of our study was a comparison of the concentration galectin-3 in patients with isolated AF without MS and patients with AF and MS, which also revealed its higher values in combination with AF and MS. The correlation of galectin-3 with the parameters characterizing obesity, the lipid profile and the level of glycemia, once again emphasizes the close relationship of this biomarker with the components of MS and enlargement of atrial volumes is also associated with an increase galectin-3 concentration, which also contributes to the risk of AF.

Galectin-3 is a key factor in the pathogenesis of fibrosis development, which induced complex mechanisms remodeling of the myocardial structure. On the one hand, it influences matrix metalloproteinases, limiting the degradation of the extracellular matrix. On the other hand, galectin-3 activates fibroblasts and enhances synthesis of the collagen I and III types that were proved earlier in several studies in animal models [19] N-terminal propeptide of procollagen I and III types are deposited directly into the myocardium in various heart diseases. In a study by Lopez et al., a relationship between blood circulating PINP and the development of left ventricle myocardial remodeling was established in patients with hypertension [20]. PIIIPN plays an important role in patients after myocardial infarction and myocardial remodeling. Its high concentration is associated with more severe heart failure and poor prognosis [21]. The association of these biomarkers with the risk of AF has been established in numerous studies previously [22,23,24], moreover, it has been established that in patients with AF in the left atrial tissue PINP predominates more and PIIINP less [25].

According to the results of our study, it was established for the first time that the concentration of PINP and PIIINP in blood plasma is higher in patients with AF in combination with MS compared with AF patients without MS and higher than in patients with MS without this arrhythmia. The correlation of these fibrosis biomarkers with obesity and dilation of the left atrium was established. The concentration of galectin-3 positively correlates with PINP and PIIINP circulating in plasma, which emphasizes the relationship of these biomarkers in the development of fibrosis. In a study of patients with AF who did not have a sufficiently effective drug control of the sinus rhythm, the mouth of the pulmonary veins was isolated to determine the indications. Assessing myocardial fibrosis is a difficult task in real clinical practice. Atrial myocardial structural remodeling can be detected by morphological examination of a biopsy or by magnetic resonance imaging (MRI). In our study, we used electroanatomical mapping in the CARTO3 system with the construction of low-voltage maps, we were able to determine the severity of fibrotic changes with respect to the total area of fibrosis. This technique has been well studied and is comparable with the results of fibrosis assessment with MRI [26]. The analysis of the data obtained allowed us to establish a positive relationship between galectin-3, PINP and PIIINP with fibrotic changes in the left atrium in patients with AF without organic heart diseases, which probably determines the pathogenetic role of these biomarkers in the development of AF. The ROC analysis allowed us to confirm the association of galectin-3, PINP and PIIINP with AF risk in the studied cohorts of patients, and to determine the threshold values of these indicators, exceeding which significantly increase the risk of this arrhythmia. It should be emphasized that in the cohort of patients with MS, it was galectin-3 that to a greater extent determined the prediction of AF risk.

The development of AF is a multifactorial process, which is based on structural and electrical changes in the myocardium, hemodynamic stress and neurohumoral effects. The processes of inflammation and fibrosis in the myocardium play an important role in structural remodeling and the appearance of micro re-entry. It is a very important identification of new biomarkers that may indirectly indicate the activity of fibrosis processes. Earlier in our study, it was found that the concentration of galectin-3 in patients with MS and no effect from drug antiarrhythmic therapy was 2.6 folds higher than in patients with effective therapy who did not have registered episodes of AF during the observation period [18]. Expanding knowledge of the pathogenesis of myocardial fibrosis induced by galectin-3 and an increase in the concentration of type I and III procollagens is important, since a high level of galectin-3, in our opinion, is a predictor of arrhythmia progression and can be considered as an additional indication for catheter radiofrequency ablation in patients with AF. This is confirmed by the fact that in recent years, many studies have been published on this problem area. In the experimental part of the work on animals, Yoshio Takemoto demonstrated the pathogenetic mechanisms of the effect of galectin-3 on the development of myocardial fibrosis and the induction of AF. The researchers also found that in patients with AF after radiofrequency ablation, a high level of galectin-3 is associated with the risk of recurrence of this arrhythmia (OR = 1.20, 95% CI = 1.01–1.45, *p* = 0.04) [27]. Clementy examined 160 patients with paroxysmal and persistent AF after radiofrequency ablation and found that a high level of galectin-3 in blood serum (OR = 1.07, 95% CI = 1.01–1.12, *p* = 0.02) and enlargement of the left atrium (OR = 1.07, 95% CI = 1.03–1.12, *p* = 0.001) were independent predictors of recurrence of AF paroxysms [28]. Thus, the association of galectin-3 with the development of myocardial fibrosis and the risk of AF development is not in doubt, especially in patients with MS. However, prospective observation and further examination are required to determine the clinical significance of this biomarker in metabolic syndrome patients.

## 4. Materials and Methods

In the period from 2014 to 2018, *n* = 1307 patients were examined with AF hospitalized in the therapeutic department of the University Hospital. The coronary heart disease was diagnosed in 721/1307 (55.2%) patients, valve pathology in 46/1307 (3.5%), 80/1307 (6.1%) patients had inflammatory heart diseases. In a further prospective study, patients of both sexes 35–60 years old were included with an isolated AF (*n* = 72) and AF in combination with MS (*n* = 176). The control groups included patients with MS without AF (*n* = 161) and healthy people without MS and AF (*n* = 71). All patients with MS had 3 or more components diagnosed according to International Diabetes Federation (IDF) criteria (2005). Patients with verified coronary heart disease, chronic heart failure, heart valve pathology, inflammatory, systemic and oncological diseases, as well as with chronic kidney disease, liver pathology with impaired function, thyroid diseases, cerebrovascular diseases, patients with a history of surgery or other heart interventions were excluded from the study. The study was approved by the Ethics Committee of Pavlov University (№456-239-A, 26.11.2013). Written informed consent was obtained from each patient.

Anthropometric and laboratory parameters were evaluated in all examined persons. The echocardiography protocol was performed in standard modes on a Vivid7 apparatus (GE Healthcare, 4855 W. Electric Ave Milwaukee, WI, USA). All biosamples of serum and plasma were centrifuged, were collected and frozen at −40 °C. The level of galectin-3 in the serum was determined by ELISA (ELISA kit, eBioscience, Vienna, Austria), allowing determination of the concentration in the range 0.47–30.0 ng/mL. The concentration of PINP and PIIINP were determined in plasma by ELISA (ELISA kit, Cloud-Clone Corp., 23603 W. Fernhurst Dr., Unit 2201, Katy, TX, USA) in the range for PINP, 33–5000 pg/mL, and for PIIINP, 2.14–400 ng/mL.

All patients with ineffective antiarrhythmic therapy and indications for interventional treatment of AF were included in a group with radiofrequency ablation. The construction of bipolar amplitude maps of the left atrium was performed under X-ray conditions using a nonfluoroscopic CARTO3 electroanatomical mapping system (Biosense Webster, Inc. 33 Technology Drive Irvine, CA, USA) and a catheter with measurement of the contact force with the left atrium myocardium (Smart Touch Thermocool (Biosense Webster, Inc. 33 Technology Drive Irvine, CA, USA). The low-voltage zones in the amplitude spectrum of 0.2–1.0 mV were estimated with their area measured using the software function of the CARTO3 navigation system “area measurement” in the off-line mode. The severity of fibrosis was estimated in the ratio of the area of fibrosis (low-voltage zone) to the total area of the left atrium and was expressed as a percentage.

All the obtained research results were included in the original database. Depending on the type of distribution, quantitative variables are represented by the mean value (M) ± standard deviation or the median (Me) with an indication of interquartile intervals (25–75%). Box plots were drawn for concentrations of biomarkers. The median (middle quartile) is shown by a line that divides the box into two parts, where 75% and 25% are the upper and lower sides of the box plot, respectively. The upper and lower whiskers represent scores outside the middle 50%. Multiple comparisons in groups (more than two) in parametric statistics were performed using one-way analysis of variance (ANOVA), and for nonparametric statistics, the Kruskal–Wallis test. The Bonferroni amendment was considered. In assessing the significance of the correlation coefficient, the Pearson criteria for the normal distribution and Spearman for the abnormal distribution of indicators were used. To assess the effect of biomarker concentrations on the probability of AF in a cohort of patients with MS, a ROC analysis was performed with the threshold value of this indicator determined. Statistical analysis was performed using IBM SPSS Statistics licensed software, version 22.0 (New York, NY, USA).

## 5. Conclusions

**1.** Galectin-3, PINP and PIIINP concentrations are higher in patients with AF and MS than in patients with AF without MS and MS without this arrhythmia.**2.** Galectin-3 serum concentration positively correlates with parameters characterizing obesity and dilatation of the atria.**3.** The severity of left atrial myocardial fibrosis is associated with an increase in the concentration of galectin-3, PINP and PIIINP.**4.** An increase in serum galectin-3, PINP, PIIINP concentration can increase the risk of AF in patients with MS.

### Limitations

Patients with AF and MS were treated with medications (antiarrhythmics, antihypertensive drugs, antithrombotics, statins), so this treatment could some extent affect the results of the study. The number of patients who underwent measurement of severity of left atrial fibrosis was not numerous, so, a larger amount of data should be collected and further monitoring of patients with MS should be performed.

## Figures and Tables

**Figure 1 ijms-21-05689-f001:**
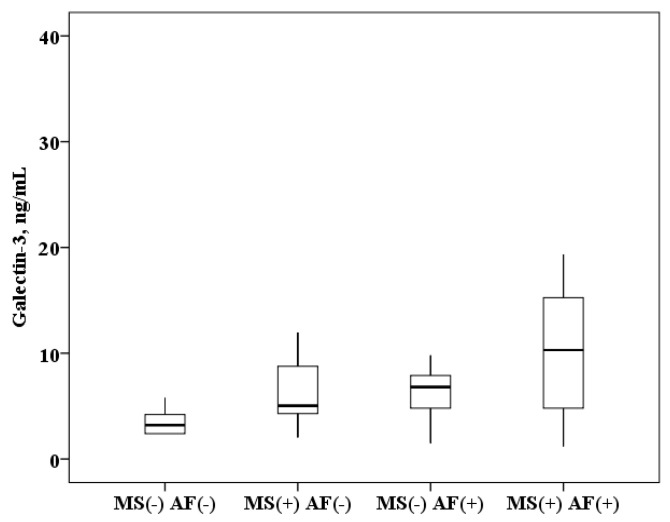
The concentration of galectin-3 in serum: healthy subjects MS(−) AF(−), patients with metabolic syndrome without atrial fibrillation MS(+) AF(−), patients with atrial fibrillation without metabolic syndrome MS(−) AF(+) and patients with atrial fibrillation with metabolic syndrome MS(+) AF(+). The median (middle quartile) is shown by a line that divides the box into two parts, where 75% and 25% are the upper and lower sides of the box plot, respectively.

**Figure 2 ijms-21-05689-f002:**
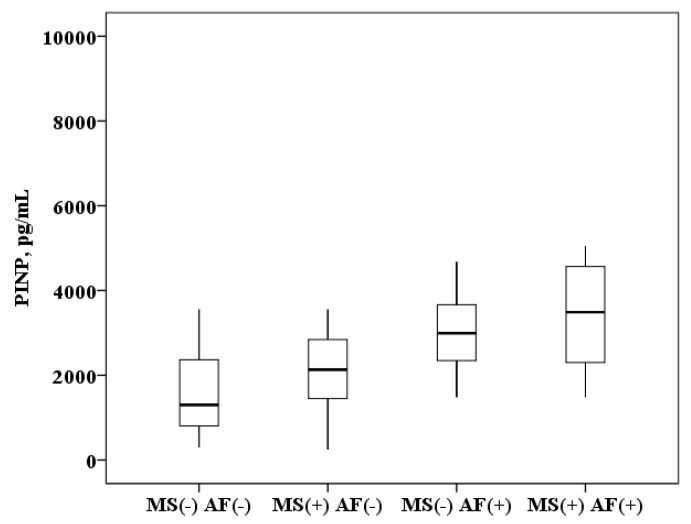
The concentration of N-terminal propeptide of collagen I type (PINP) in the plasma: healthy subjects MS(−) AF(−), patients with metabolic syndrome without atrial fibrillation MS(+) AF(−), patients with atrial fibrillation without metabolic syndrome MS(−) AF(+) and patients with atrial fibrillation with metabolic syndrome MS(+) AF(+). The median (middle quartile) is shown by a line that divides the box into two parts, where 75% and 25% are the upper and lower sides of the box plot, respectively.

**Figure 3 ijms-21-05689-f003:**
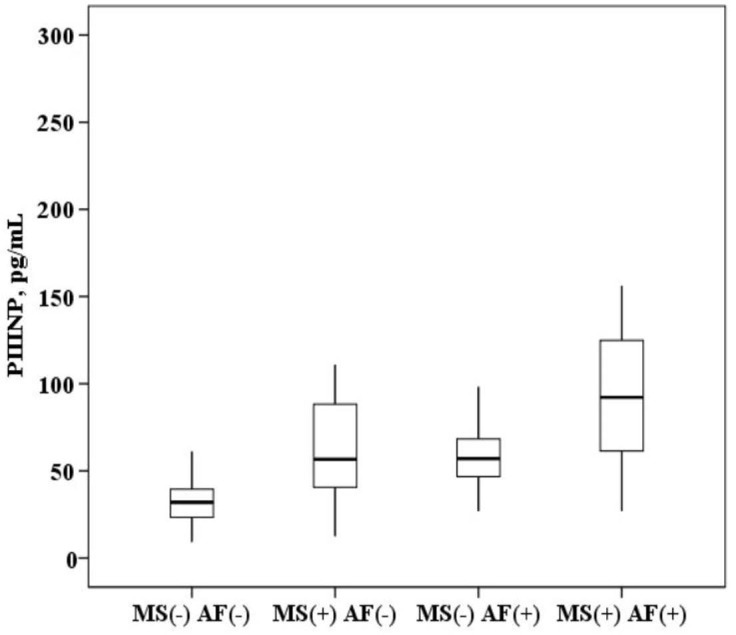
The concentration of N-terminal propeptide of collagen III type (PIIINP) in the plasma: healthy subjects MS(−) AF(−), patients with metabolic syndrome without atrial fibrillation MS(+) AF(−), patients with atrial fibrillation without metabolic syndrome MS(−) AF(+) and patients with atrial fibrillation with metabolic syndrome MS(+) AF(+). The median (middle quartile) is shown by a line that divides the box into two parts, where 75% and 25% are the upper and lower sides of the box plot, respectively.

**Figure 4 ijms-21-05689-f004:**
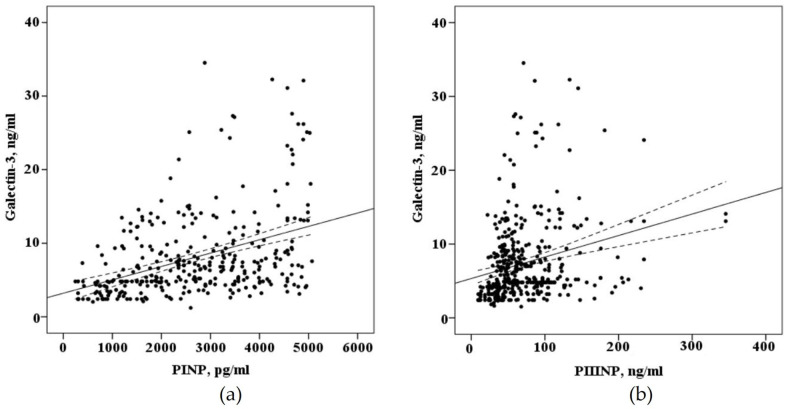
The correlation between the blood concentration of biomarkers fibrosis with interpolation line and 95% confidence interval: (**a**) correlation between the blood concentration of galectin-3 and N-terminal propeptide of collagen I type (PINP); (**b**) correlation between the blood concentration of galectin-3 and N-terminal propeptide of collagen III type (PIIINP).

**Table 1 ijms-21-05689-t001:** Clinical, laboratory and echocardiographic characteristics of the investigated groups.

Parameters		MS (−) AF (−) *n* = 71 (1)	MS (+) AF (−) *n* = 161 (2)	MS (−) AF (+) *n* = 72 (3)	MS (+) AF (+) *n* = 176 (4)	*p*
Anthropometric and laboratory parameters						
Age, years		51.3 ± 8.6	53.7 ± 9.3	55.6 ± 6.8	54.3 ± 7.2	*p* > 0.05
Sex, male/woman		34/37	94/67	30/42	98/78	*p* > 0.05
BMI, kg/m^2^		22.5 ± 4.8	34.1 ± 8.6	25.9 ± 3.5	30.3 ± 6.6	*p*_1,2_ < 0.001; *p*_1,3_ = 0.456; *p*_1,4_ < 0.001 *p*_2,3_ < 0.001; *p*_2,4_ = 0.134; *p*_3,4_ < 0.001
Waist circumference, cm		79.5 ± 8.1	114.8 ± 11.5	82.7 ± 11.7	111.9 ± 13.5	*p*_1,2_ < 0.001; *p*_1,3_ = 0.578; *p*_1,4_ < 0.001 *p*_2,3_ < 0.001; *p*_2,4_=0.423; *p*_3,4_ < 0.001
Total cholesterol, mmol/L		4.9 ± 0.9	5.4 ± 1.1	4.8 ± 1.2	5.2 ± 1.2	*p*_1,2_ < 0.001; *p*_1,3_ < 0.878; *p*_1,4_ < 0.001 *p*_2,3_ < 0.001; *p*_2,4_ < 0.345; *p*_3,4_ < 0.001
LDL-cholesterol, mmol/L		2.8 ± 0.3	3.4 ± 0.3	3.1 ± 0.3	3.1 ± 0.4	*p*_1,2_ < 0.001; *p*_1,3v_< 0.001; *p*_1,4_ < 0.001 *p*_2,3_ = 0.567; *p*_2,4_ = 0.675; *p*_3,4_ = 0.961
HDL-cholesterol, mmol/L		1.6 ± 0.3	1.2 ± 0.3	1.4 ± 0.3	1.3 ± 0.4	*p*_1,2_ < 0.001; *p*_1,3_ = 0.678; *p*_1,4_ < 0.001 *p*_2,3_ < 0.001; *p*_2,4_ = 0.871; *p*_3,4_ < 0.001
Triglycerides, mmol/L		1.0 ± 0.3	2.1 ± 0.8	1.3 ± 0.4	1.7 ± 1.2	*p*_1,2_ < 0.001; *p*_1,3_ = 0.341; *p*_1,4_ < 0.001 *p*_2,3_ < 0.001; *p*_2,4_ = 0.121; *p*_3,4_ < 0.001
Glucose, mmol/L		4.7 ± 0.6	6.1 ± 1.2	5.1 ± 0.4	6.0 ± 1.4	*p*_1,2_ < 0.001; *p*_1,3_ = 0.068; *p*_1,4_ < 0.001 *p*_2,3_ < 0.001; *p*_2,4_ = 0.891; *p*_3,4_ < 0.001
Echocardiography						
Left atrium volume, mL		43.2 ± 9.4	81.9 ± 16.6	60.4 ± 19.8	79.9 ± 19.4	*p*_1,2_ < 0.001; *p*_1,3_ < 0.001; *p*_1,4_ < 0.001 *p*_2,3_ = 0.01; *p*_2,4_ = 0.124; *p*_3,4_ = 0.01
Left atrium volume index, mL/m^2^		24.3 ± 4.9	39.2 ± 9.7	30.4 ± 9.0	40.1 ± 11.2	*p*_1,2_ < 0.001; *p*_1,3_ < 0.001; *p*_1,4_ < 0.001 *p*_2,3_ = 0.01; *p*_2,4_ = 0.227; *p*_3,4_ = 0.01
Right atrium volume, mL		41.3 ± 8.9	68.5 ± 14.4	57.5 ± 20.6	65.9 ± 14.7	*p*_1,2_ < 0.001; *p*_1,3_ < 0.001; *p*_1,4_ < 0.001 *p*_2,3_ = 0.01; *p*_2,4_ = 0.136; *p*_3,4_ = 0.01
Right atrium volume index, mL/m^2^		23.4 ± 4.3	31.9 ± 7.3	29.2 ± 8.8	32.8 ± 7.8	*p*_1,2_ < 0.001; *p*_1,3_ < 0.001; *p*_1,4_ < 0.001 *p*_2,3_ = 0.01; *p*_2,4_ = 0.136; *p*_3,4_ = 0.01
EF, %		64.3 ± 7	61.2 ± 6	62.4 ± 4.2	60.8 ± 6	*p* > 0.05
AF duration, years		-	-	4.9 ± 1.2	4.2 ± 2.2	*p* > 0.05
AF form	Paroxysmal	-	-	40/72 (55.6%)	90/176 (51.1%)	*p* > 0.05
	Persistent	-	-	32/72 (44.4%)	86/176 (48.9%)	*p* > 0.05
Pharmacotherapy						
ACEi, ARBs		-	106/161 (65.8%)	12/72 (16.7%)	145/176 (82.3%)	*p*_2,3_ = 0.01; *p*_2,4_ = 0.03; *p*_3,4_ = 0.01
Beta blockers		-	89/161 (55.3%)	53/72 (73.6%)	137/176 (77.8%)	*p*_2,3_ = 0.01; *p*_2,4_ = 0.01; *p*_3,4_ > 0.05
Diuretics		-	86/161 (53.4%)	7/72 (9.7%)	85/176 (48.3%)	*p*_2,3_ = 0.01; *p*_2,4_ = 0.02; *p*_3,4_ < 0.001
Ca-channel blockers		-	76/161 (47.2%)	-	95/176 (53.9%)	*p*_2,4_ > 0.05
Statins		-	98/161 (60.7%)	8/72 (11.1%)	117/176 (66.5%)	*p*_2,3_ = 0.01; *p*_2,4_ > 0.05; *p*_3,4_ < 0.001
Antiarrhythmic therapy		-	-	19/72 (26.4%)	39/176 (22.2%)	*p*_3,4_ > 0.05
Antiplatelet agents		-	36/161 (22.4%)	2/72 (2.8%)	5/176 (2.8%)	*p*_2,3_ = 0.001; *p*_2,4_ = 0.001; *p*_3,4_ > 0.05
Anticoagulants		-	-	28/72 (38.9%)	135/176 (76.7%)	*p*_3,4_ < 0.001

ACEi—angiotensin-converting enzyme inhibitors, AF—atrial fibrillation, ARBs—angiotensin II receptor blockers, BMI—body mass index, EF—left ventricular ejection fraction, HDL—high-density lipoproteins, LDL—low-density lipoproteins, MS—metabolic syndrome.

**Table 2 ijms-21-05689-t002:** Results of receiver operating characteristic (ROC) analyses of AF risk stratification in the total cohort of patients and in a cohort of patients with metabolic syndrome (MS).

Biomarkers	AUC ± SD	95% CI	*p*	Cut-Off Value of Biomarkers	Risk of AF(OR)	95% CI	*p*
The total cohort of patients							
Galectin-3(1)	0.768 ± 0.03	0.717–0.819	*p*_1,2_ = 0.415 *p*_1,3_ < 0.0001 *p*_2,3_ < 0.0001	5.8 ng/mL	4.8	3.3–7.01	<0.0001
PINP(2)	0.761 ± 0.03	0.708–0.813		2896.8 ng/mL	5.6	3.5–9.1	<0.0001
PIIINP(3)	0.707 ± 0.03	0.651–0.763		45.8 pg/mL	7.1	4.3–11.5	<0.0001
The cohort of patients with metabolic syndrome (MS)							
Galectin-3(1)	0.788 ± 0.03	0.727–0.847	*p*_1,2_ = 0.023 *p*_1,3_ < 0.0001 *p*_2,3_ < 0.0001	12.6 ng/mL	5.3	3.1–9.1	<0.0001
PINP(2)	0.770 ± 0.03	0.706–0.834		3426.8 ng/mL	8.9	4.8–6.9	<0.0001
PIIINP(3)	0.718 ± 0.03	0.651–0.784		56.7 pg/mL	4.4	2.6–7.5	<0.0001

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
