# Peer review of "Galectin-3, N-terminal Propeptides of Type I and III Procollagen in Patients with Atrial Fibrillation and Metabolic Syndrome"

_ijms, 2020, doi:10.3390/ijms21165689_

Round 1

Reviewer 1 Report

The introduction mainly describes galactin 3. Please also describe PINP and PIIINP proportionally

Table 1 presents the results of BMI, WC, Total cholesterol, LDL in patients with AF, MS. Please show the correlations of risk factors with levels of galactin-3, PINP, PIIINP. It seems very interestingIn the material and methods section please  the criteriafor inclusion and exlusion of patients we know that some diseases as inflamation or cancer can affect the amount of serum concentrations of the tested proteins Figures 1,2,3, please describe exactly whether the lines in the charts mean average or medianIn the discussion, I am asking for a more detailed description of the clinical applications of the tested proteins, in what group of patients and in what situations

Author Response

Thank you for reviewing our publication! Your professional opinion is very important for our scientific team and we tried to make the maximum adjustments to your questions and suggestions! You can appreciate the changes in the attached file and the original manuscript.

Reviewer 2 Report

This manuscript provides information on the circulating levels of galactin - 3 and N-terminal propeptides of type I and III procollagen in patients with atrial fibrillation and metabolic syndrome.  The authors have described an increase in the levels of these biomarkers with interesting conclusions. The manuscript requires revision in terms of the adherance to the instructions to the authors and formatting. The following editorial recomendations are provided to revise this manuscript:

  1. The manuscript is written in poor English with several spelling and typographic errors. It should be carefully checked. 
  2. Some the abbreviations used are without any description. All abbreviations should be explained completely when used initially.
  3. Table 1 should be accompanied with a clear legend with the abbreviation. Additioanlly multiple p values are given without explanation in some of the columns
  4. Figures 1 - 4 should be accompanied with more extended legends
  5. There should be a final section on the conclusions

Author Response

(The authors gave the same response as above.)
